# A Psychology of Sustainable Career Development: Hypernormalized Ideology or Inherently Sustainable?

**P. Matthijs Bal \***  **and Roxana Alhnaity**

Lincoln International Business School, University of Lincoln, Lincoln LN6 7TS, UK; ralhnaity@lincoln.ac.uk
\* Correspondence: mbal@lincoln.ac.uk

**Abstract:** Sustainable career development is a great priority for organizations, governments and individuals alike. Facing the grand challenges of our global world, careers and their development have to be re-designed to incorporate more sustainable ways of living and working. However, most work around sustainable careers is centered around neoliberal modes of organizing, amplifying individual responsibility of individuals for their careers, while treating careers merely as an instrumental 'tool' towards organizational performance and viability. Hence, sustainable careers are a hypernormalized ideology. In the current paper, a psychology of sustainable career development is introduced that deviates from earlier, more conservative models, of career development towards a more radical interpretation and recognition of truly sustainable ways of organizing and developing careers. Anchored in an interpretation of sustainable careers as promoting dignity and capabilities of people, this conceptual paper formulates a new psychology of the sustainable career, towards integration rather than individualization.

**Keywords:** sustainable careers; ideology; hypernormalization; dignity; psychology

## 1. Introduction

Contemporary careers in and across organizations are no longer supposed to be linear, stable and secure, but are increasingly required to be 'sustainable' [1,2]. Individuals in the contemporary economy are therefore no longer expected to be focused on having a stable job and continued employment across one's career, but to be an 'entrepreneur of the self' [3]. In the current neoliberal world, the 'flexible human being' has become fetishized, projecting a worldview of the contemporary individual as being agentic, proactive, flexible and capable of managing and steering one's own life. Human beings are no longer merely the product of their circumstances, either genetic or social, but are expected to take matters into their own hands and lead their lives as healthy, happy and productive citizens and workers [4,5]. Absent from such notions are deliberations around what this 'productivity' requirement entails, why it has become so central to projections about the contemporary human being and how the proposed benefits to be 'productive' materialize for individuals in terms of development of a sustainable career. Nonetheless, it is posed as a sine-qua-non for the contemporary worker, and similar descriptions have been found in relation to sustainable careers [4]. Sustainable careers are also a function of agentic behaviors, individual responsibility and flexibility, while at the same time supposedly ensuring happy, healthy and productive workers [4]. The intellectual underdevelopment underlying such models requires a one-dimensional supportive empirical approach: cases and statistics are sought to support such models, whereby successful employees who have gained sustainable careers are successful because they have been flexible and agentic, while the absence of such actions and attitudes cause individuals to miss out on the possibility to build a sustainable career [6]. There is no critical approach towards these models, and thus scholarship only looks for support for these models while ignoring the contexts and causes for the absence of sustainable careers. For instance, when individuals can only have

a sustainable career when they are agentic, scholars do not question *why* agentic behaviors are absent or impossible within many contexts. Hence, the logic behind sustainable careers research is onefold and without complexity: individuals in the contemporary workplace need to be flexible and agentic so that they can have a sustainable career, which is denoted by an individual being 'happy, healthy and productive'. Those who refuse or have no option to be flexible are excluded from having a sustainable career. The simplicity of such theorizing dictates an approach that is prone to confirmation bias: positive examples of sustainable careers are looked after, while reasons for the absence of sustainable careers are neglected.

This also points to the exclusionary nature of sustainable careers. Bal and colleagues [6] analyzed *who* and *what* are excluded from sustainable careers. A sustainable career is exclusively conceptualized around the possibility for an individual to be a productive citizen in neoliberal society and to play their role as loyal citizens fulfilling their 'productive duty': to be employed throughout one's career, to refrain from long-term absence from work (e.g., because of illness, voluntary work or other priorities such as caring responsibilities) and to be perpetually flexible in light of a rapid changing society. However, lines of exclusion cross along privilege (i.e., only privileged individuals obtain sustainable careers) and anthropocentrism (i.e., economic logic, by definition, trumps concern for the environment and the planet). These lines of exclusion delegitimize the very concept of sustainable careers as any use of the term manifests at the expense of those unable to obtain a sustainable career, including those who are unable to express their voice (e.g., animals, nature). In the following paper, we will develop a psychology of sustainable careers, where we will introduce a dignity-inspired conceptualization of sustainable careers that is fundamentally opposed to existing conceptualizations (e.g., [4]). The objective of the paper is to provide a novel perspective on sustainable career development that promotes dignity of people and the planet and therefore moves beyond existing hegemonic instrumental and individualistic notions of sustainable career development. In so doing, this paper offers an attempt to 'rescue' the concept from its inherent neoliberalization and hypernormalization, towards an understanding of sustainable careers that have a value not just for privileged individuals, but for the most vulnerable on our planet, including people, animals and the environment. The contributions of this paper include a refocusing of sustainable careers in light of the need for dignity, capabilities and equal rights for human beings and the planet. It is not sufficient to rely upon individual agency to achieve sustainable careers, and governmental and organizational support for sustainable careers is needed, yet inherently risks becoming neoliberalized when perceived as instrumental, even if such instrumentality aims for individual flourishing, organizational viability or public health. In such cases, instrumentality dictates a utility, thereby emptying concepts from their inherent properties towards a means towards an end. This paper, therefore, will radically deviate from such understanding to provide a more fundamental reassessment of the concept and its potential value in contemporary, global society. While there has been critical literature in career counseling (e.g., [7,8]), this is still largely missing from the sustainable careers literature, and thus in this paper we contribute to more critical understandings of sustainable careers.

## 2. Sustainable Careers as Hypernormalized Ideology

While the concept of sustainable careers has received growing attention in the literature, as evidenced by handbooks and special issues [1,4], it has also been criticized [6]. A critical aspect of sustainable careers pertains to its inherently privileged, primarily Western perspective. For instance, the notion of a 'sustainable career' remains somewhat ludicrous for someone in the Bangladeshi RMG sector, despite the inherent needs for more sustainable approaches to the manufacturing of clothes and the labor conditions in this industry [9]. The question, therefore, is, who is being referred to when scholars write about sustainable careers? Usually, examples and inherent references are made to privileged, highly educated, (Western) workers who already display many of the characteristics needed to individually arrange a sustainable career (i.e., being flexible and agentic). However, this population is

only a fraction of the world's entire population, making the concept redundant for the large majority of people in the world. Why, then, does this blind spot among scholars working on this exist? This not only signifies their own privileged positions within (Western) universities that make them more likely to sympathize with workers who are just like them, but also the primary focus of this work, which is not supposed to have real meaning beyond the context in which the work is produced. With the aim of educating privileged groups in society, teaching and research within business schools are usually focused on the needs of such groups, without considering that, for the vast majority of the people on this planet, such concepts have little to no meaning. What is the meaning of such work when it, by definition, excludes most people in this world?

A first assessment must be made around the ideological nature of sustainable careers [6]. While not everything is ideological per se, everything can be studied from an ideological perspective [10,11]. In other words, the sustainable careers concept does not necessarily have to be ideological but can be studied and interpreted from an ideological perspective. While Bal et al. [6] analyzed the concept from a neoliberal-ideological perspective, elucidating the inherent links between the conceptualization of sustainable careers as driven by individualism, agentism and flexibility and neoliberal ideology (see e.g., [12]), the current paper does not aim to repeat such analysis but primarily focuses on the performative effects of such ideology. In other words, even though unmasking of the neoliberal ideology behind sustainable careers may be informative, it does not reduce the performative effects of the original conceptualization of sustainable careers; it remains effective in both science and practice. In scientific research, an uncritical adoption of sustainable careers as defined hegemonically (i.e., flexible individuals proactively ensuring their own sustainable careers) remains the norm, and perhaps might even be favored through the publication process itself. Critical perspectives tend to be ignored or rejected as editors and reviewers may likely be those with vested interests into a status-quo treatment of the concept, while approaches that uncritically comply with dominant theorizing more easily get through the hierarchical academic system (i.e., established scholars act as gatekeepers within the system, and thereby allowing certain perspectives, while excluding other perspectives and voices from the academic field).

Through this process, the sustainable careers concept becomes 'hypernormalized ideology' [13]; it is not merely a concept that can be understood through ideological analysis [6], but additionally, it becomes 'hypernormalized' [14]. Hypernormalization was originally coined by the Russian anthropologist Alexei Yurchak in describing the gap between authoritative discourse and really existing practice in the late Soviet Union after the death of Stalin (post 1953). While the Soviet rulers maintained a strict compliance with the acceptable discourse under Stalin, the practice of everyday reality was opposed to such discourse, as evidenced by the terror practiced in the Soviet Union under the retention of slogans such as collectivity and solidarity. While this gap grew to absurd proportions in the last decades of the Soviet Union until its collapse in the late 1980s and early 1990s, it remained 'hypernormalized', or treated as entirely normal, taken for granted, and the accepted state of affairs. Yurchak's work showed how despite the absurdity of the gap between authoritative discourse and practice, people continued to believe and invest in authoritative discourse and the ideals of collectivity, solidarity and brotherhood under a Communist system. Hence, even though discourse was separated from reality, it remained to have an impact on individuals, and was internalized into the core beliefs and fantasies of people about the ideal state. In a similar vein, sustainable careers are also hypernormalized; even though the conceptualization of the concept becomes 'absurd' [13] in its exclusionary nature describing only the attitudes and behaviors of a very small number of hyperprivileged individuals in the world, it remains performative through its internalization into the core beliefs of people that they should exhibit the necessary traits in order to obtain a sustainable career, and thereby to be healthy, happy and productive. Even though the concept of a sustainable career may be absurd to the vast majority of the world's population, it still has a performative effect, even beyond self-internalization of individuals that they should be

agentic, proactive and flexible in order to live a meaningful life. As performativity remains poorly contested, governments and organizations alike uncritically adopt concepts such as sustainable careers into policy making (e.g., ILO and the UN adopting concepts such as 'green, sustainable jobs' without any evidence of actually materializing jobs that contribute to greater sustainability). The effect of this is that concepts that have an inherent ideological connotation are used nonetheless as an 'empty signifier' [15]; in their lack of meaning, they can be filled implicitly ideologically, while at the same time being treated as something inherently good or positive that everyone could or should rally behind. Meanwhile, in their emptiness, their performative effects on propagating a specific understanding of sustainable careers remain effective, whereby critical perspectives are suppressed even more strongly. This shows again how dominant ideological understandings of neoliberal sustainable careers become hypernormalized and, as such taken for granted, treated as entirely normal and accepted without discussion. The question, then, is whether the concept has a potential beyond this neoliberal understanding. To do so, we need to formulate a psychology of the sustainable career and to consider the lines of exclusion (i.e., who and what is excluded from sustainable careers).

## 3. Introducing a Psychology of the Sustainable Career

Thus far, sustainable careers have merely been treated as an organizational concept; they are supposed to be of primary benefit to organizations (i.e., careers are only sustainable when they are associated with productivity [4]. There is no psychology of sustainable careers, as only some attitudinal and behavioral tendencies are related to sustainable careers (e.g., proactivity, flexibility). To do so, we will formulate a psychology of sustainable careers based on the identification of the basic human needs throughout their lives, using both a dignity lens [13] and a capability approach [16]. These theories postulate that in the face of the fundamental absurdity and meaninglessness of human life [13], people search for their dignity to be respected and protected while they look for the opportunity to exercise their capabilities. While the sustainable careers literature maintains that careers are only sustainable when they are defined by agency and meaning, an opposed perspective is presented by Bal et al. [13] in their theory of the absurd workplace; on the basis of their analysis of the contemporary world and workplace as absurd, lack and loss of meaning is central to organizing, and, as a result of the dissolution of proposed meaning into the absence of meaning, the illogical and the absurd define the state of our contemporary world. It is no surprise, then, to observe a rise in experienced burnouts, as this represents the maladaptive responses to an inherently alienating and absurd system and workplace.

One of the core problems around the use of 'sustainable' in relation to careers is not the notion that careers are limited in scope and cannot be described sensibly over the course of many years without creating some post-hoc rationalization of the randomness of choice and human decision-making, but that 'sustainable' is used all too freely. Despite it being an empty concept, it has the connotation of something inherently desirable, which masks its exercise of biopower, control and symbolic violence projected upon individuals who are expected to have a 'sustainable career' in an increasingly unstable world, while being blamed for not surviving and thriving in the disintegrating contemporary capitalist system. Hence, the contemporary existence of 'career' is deeply traumatic for most people globally in the absence of any kind of sustainability, which makes the sustainable career a fantasy construction [6] that functions more as a projection (i.e., as something to fantasize about rather than being rooted firmly in the everyday experience of the majority of people). Using the term obfuscates the more traumatic reality that 'cannot be symbolized' in public discourse [17]. In other words, the reality of unsustainable careers (or total lack of any career prospects) is denied through the projection of the sustainable career on organizational reality as well as on policy frameworks. The sustainable career thereby functions as authoritative discourse, which may be fundamentally opposed to reality. After all, how many people on this planet will truly experience their careers as sustainable, and not subject to volatility, turbulence and rapidly changing societal and

organizational realities, and has this not been the case throughout history? The notion of a stable career (i.e., the lifetime employment model of post-war social democracies) functioned as an anomaly, and the current fundamentally unstable reality is the actual 'normal'. Hence, rather than speaking of the sustainable career as ensuring ontological security [18], it is more appropriate to recognize the fundamental ontological insecurity of the contemporary career. While the Western world has broken down its social democracy, including its protective framework, to a void neoliberal state, there never was such social democracy in most of the world (or only for brief spells). Consequently, the notion of a 'career' always had limited applicability to most of the world's nations and would be driven primarily by a need for (ontological) security, that is, a security about one's identity and sense of self, which is perpetually endangered as a function of societal unrest and global upheaval. Hence, the notion of sustainable careers is hypernormalized, and would better be understood from the psychological perspective of dignity and capabilities. In so doing, it is about recognizing the inherent unsustainability of our contemporary ways of living, which are hypernormalized (i.e., the Western, neocolonial, patriarchal, capitalist mode of existence that continuously threatens the sustainability of the planet [13], and the resulting traumatic impact on the individual. This trauma of a broken planet that cannot be healed easily through (collective) human agency (which constitutes a fundament of human existence) has been rather ignored in the literature and only received attention during the last years, as evidenced by research on climate trauma or climate grief [19,20].

The possibility of a stable sustainable career therefore constitutes an absurdity in the face of climate destruction and associated with it global migration, wars, poverty and inequalities [13]. This reflects not just a collective understanding concerning the predicament of the world but can also be understood as an individual psychological response: there is either disavowal or trauma response. On the one hand, a psychology of the sustainable career holds the possibility of cynical disavowal [17]. In this case, the individual denies the traumatic reality of our profoundly unsustainable and unstable societies and lifestyles and becomes the neoliberal citizen by focusing on one's own outcomes, utility and self-centered behaviors to ensure a sustainable career in the rather narrow way defined conceptually (e.g., to be agentic, proactive and flexible to sustain one's own individual health and career success). The philosopher Žižek [17] refers to this process as disavowal or reversing the well-known Marxist formula of 'Sie Wissen nicht was Sie tun, aber Sie tun es' into 'they very well know what they are doing, but nonetheless are doing it'. In other words, there is an 'enlightenment': people are (even though at times at a very basic level) aware of the unstable global society surrounding them, which delivers news of wars, climate destruction, poverty and inequality on a daily basis, but nonetheless they pretend as if all is normal [13]. The hypernormalization of the absurd becomes cynically disavowed, such that people can cope with the traumatic realities by denying them, and to hold on to privileges that have been gained or inherent to one's make up as a human being (e.g., being white). In this way, those people who, because of their privileges already had access to sustainable careers, simply deny reality in favor of a more fantasmatic symbolic reality (i.e., that they have obtained their sustainable careers because they worked hard for it, despite all the others worldwide who worked just as hard or harder but never had any possibility for a career at all).

On the other hand, those who have experienced their 'absurd moment' [13] may see through the smokescreen of authoritative discourse, opening up the possibility for 'seeing reality as it really is'. The absurd moment occurs when an individual becomes aware of the meaninglessness of life, discourse and social practices. This unmasking or exposing of a more traumatic reality behind the symbolic features of society is a breaking through hypernormalization, the absurd moment that was linked already by philosopher Albert Camus to feelings of despair, anxiety and a total loss of hope [13]. It represents the moment when the void is opened up and an individual asks the 'why' of a meaning system [21]. It is that moment when an individual realizes that despite any rhetoric of the actually achievable sustainable career, there is no such possibility, and one is confronted with the Lacanian Real,

or the traumatic kernel that cannot be symbolized [17]. This Real is the void, that which cannot be captured through authoritative discourse, that which falls outside the scope of the agentic, proactive, flexible human being, and in fact refers to actually existing people and their experienced lives. These people are alienated from a patriarchal, neocolonial, capitalist global system and experience the trauma of this alienation, the existential separation from the Other, which could also be referred to as the well-known experience of burnout. Recent research on climate trauma, grief and anxiety underpin the relevance of these absurd moments and their effects, especially for younger generations [19]. These are moments where individuals realize that all securities and certainties of our contemporary world are all but certain, and such moments may cause great stress and anxiety. While there may still be ideological and libidinal investment into a fantasized sustainable career, its effective functioning ceases to be operative, and cracks emerge within the perception of reality and the void behind this reality. The absurd moment dictates that careers are unsustainable and that sustainability as a concept is hypernormalized; while it is treated as something that belongs to the aspirational level of (Western) countries, the world as a whole is on another path and does not treat sustainability as a truly meaningful anchor for political decision making. Unsustainable human behavior is the default, and psychologically this is processed by individuals not through scientific analysis, amplifying the hopelessness of our predicament, but through investment, either in the fantasized status-quo or in some uncertain alternative while trying to survive and postulating one's marginalized position within the current economy as a 'career', which in fact is nothing more than a sequence of random jobs and opportunities granted or forced through external circumstances. At the moment, there are over 108 million individuals who have been forced to leave their countries because of climate change, conflicts or other reasons [22]. Referring to their migration as a search for a sustainable career is absurd, and the question, therefore, is how to formulate and conceptualize a more inclusive sustainable career.

## 4. Inclusive Sustainable Careers

An inclusive sustainable career needs to be defined from a non-hegemonic Western perspective, and therefore we dissociate from conceptualizing agency, proactivity and flexibility as the building blocks of a sustainable career. In contrast, this paper offers an understanding of sustainable careers as conceptualized through dignity and capabilities. When sustainable careers would be implemented in scientific research, practice and policy, they need to have a more universal applicability and therefore need to be conceptualized at the level relevant for people across the world, and not merely for a small minority of hyperprivileged individuals. To do so, sustainable careers can be theorized from a dignity approach [23,24] and a capabilities approach [16].

*Dignity and Sustainable Careers*

First, inclusive sustainable careers can be theorized as respecting, protecting and promoting the dignity of individuals [13]. Dignity refers to the inherent and intrinsic worth of each individual human being on this planet. Dignity is an inalienable human right, and every human being has the fundamental right for their dignity to be respected, protected and promoted. However, the right to dignity does not yet stipulate the duty to respect dignity, and it is therefore that contemporary theory on dignity has emphasized the importance of the relational underpinning of dignity; it is through the duty to respect others' dignity and the act of recognition dignity that it manifests in social interaction [13,25]. While dignity has been somewhat absent from the organizational literature, recent work has emphasized the importance of bringing dignity to organizations [26,27]. Especially in a sphere of life that has traditionally been designed around a utilitarian philosophy, the introduction of a deontological, intrinsic philosophy of dignity brings new perspectives. In this instance, dignity offers the centrality of the human being to the concept of organizing, and thus a move away from the instrumental reasoning behind work organizations. In

this spirit, sustainable careers can be theorized accordingly by distinguishing various dimensions through which dignity informs sustainable careers.

The first major implication of a dignity approach to sustainable careers is that it is not just those individuals who are flexible, proactive and agentic who 'deserve' to have a sustainable career. In the spirit of dignity, respect for each individual human being means that no individuals are prioritized over others merely because they exert some traits favorable over others. Currently, people who are not agentic, proactive and flexible are blamed when they do not secure a sustainable career and when they are not 'happy, healthy and productive'. A dignity paradigm dictates that no matter the traits of an individual, no matter what background, race, gender, health status or any other characteristic, every individual should have the same right and access to a sustainable career. Dignity implies equality of all human beings across this world but does not presume equal standing; people differ substantially in terms of their access to resources, capabilities and innate abilities to live a dignified life. While institutional structures are generally (still) much stronger and more developed in the Western world, enhancing the possibilities for a dignified life, this is much less the case in many other countries across the world. Beyond this distinction, neocoloniality dictates the chances for dignity: is it not *because* of centuries of colonial oppression and exploitation that Western countries have been able to benefit from cheap or free access to valuable resources at the expense of the rest of the world, and yet they continue to do so. Hence, dignity never exists in a vacuum, but is always embedded in historical notions of colonial pasts, a past that is too easily denied in the West to disavow a historical guilt that still influences present day relationships and contexts. Dignity, therefore, does not merely focus on the possibility for people to live their lives in a dignified way, but also is about restoration of past violations; the dignity model of Bal [13] describes how at the very bottom level of organizational life, dignity violations ought to be prevented, while organizations can move towards more active forms of respecting and protecting dignity, creating dignified cultures and, by extension, promote dignity, including the restoration of places and contexts where dignity has been violated. Restoration, repair and healing remain the essential parts of dignity manifestation in organizations and society.

Translating such notions to the concept of sustainable careers implies that a conceptualization of a career that is sustainable includes a focus on the dignity of the individual, as well as the dignity of others involved. For instance, the conceptual paper of De Vos et al. [4] presents three vignettes of people who have successfully obtained/developed their sustainable careers. One of them is 'Emma', who works for a 'management consulting firm', which could, for instance, be Boston Consulting Group or one of the 'Big Four' accounting/consulting firms. Treating work in such companies as merely neutral and offering the possibility for a 'sustainable career' neglects the more fundamentally shady nature of the consulting industry in perpetuating a neoliberal capitalist system, in which *everything* is permitted for shareholder value and profit generation, including facilitating profits to be offshored to tax havens, lobbying governments with misinformation to block regulation of the finance industry and providing consultancy to firms as to how to make employees redundant in the cheapest way possible (see e.g., [28]). Describing such work environments as 'challenging, young and dynamic' and 'work hard, play hard' [4] does not just seem inappropriate and unreflective, but is precisely the reason why dignity is *absent* in hegemonic work on careers. Excluding a broader contextual analysis in careers research may enable the identification of a career as more or less sustainable, but utterly ignores the vignette of Emma as being a hyperprivileged individual who earns her grand salary *at the expense of* society, the most vulnerable people and, in a wider sense, the planet as a whole. Treating consulting firms as 'neutral' entities that have no role in the perpetuation of exploitative neoliberal capitalism seems a rather deliberate choice than mere ignorance.

Sustainable careers, therefore, only exist if they allow access for people worldwide and when they consider the wider costs and impacts of the accumulation of resources to ensure sustainable careers. While global access to sustainable careers can be perceived as utopian (i.e., in the same way calls for a 'universal basic income' are limited by the extent

to which it will never materialize in reality), a 'golden rule' pertains to the notion that a career can *only* be sustainable if it does not impede someone else's career sustainability. If the core business model of a firm is to exploit other people or the planet, work in these firms cannot be referred to as sustainable. In this vein, working in a fossil fuel company can never be 'sustainable' as the core business model is to exploit the earth (in a literal way) to create financial profit. While employees in these sectors may enrich themselves and may self-experience having a sustainable career, the limitation of the meaning of sustainability elucidates a second meaning of dignity: dignity is not just about the inherent, inviolable worth of human beings, but also pertains to the dignity of the planet as such [13]. As humans we do not merely live on this planet but we are dependent on the planet for our own survival. The planet and our environment provide the very conditions that sustain or limit life itself. While careers may have never been conceptualized in relation to the core fundament of human existence (i.e., the very planet we live on that sustains our existence), a career can *only* be sustainable when careers in organizations do not threaten the very sustainability of our planet. However, it is all too well known that the capitalist mode of living, with its necessity of perpetual growth, exploits the planet and makes all planetary resources (e.g., animal life, trees, the land, minerals, etc.) instrumental to the capitalist machine [15], the process through which the Western countries can provide their 'sustainable careers'. Actually existing sustainable careers, therefore, are careers that respect, protect and promote the dignity of the planet in the widest sense, including the dignity of animals, trees, plants, the land and any natural resources on the planet.

The question, then, becomes *how* careers can become more sustainable if they must be designed and developed around these conceptualizations of dignity? To do so, we build on a model closely related to dignity work [23,26], which revolves around the notion of 'capabilities' as conceptualized by Sen and further developed by Nussbaum [16,29,30]. Capabilities have not yet been linked to sustainable careers, which is somewhat surprising given the straightforward link between the possibility for a human being to exercise their capabilities and the translation of this exercise into the experience of a sustainable career. While research has prioritized an individualistic, neoliberal understanding of sustainable careers, a capabilities approach to understand sustainable careers indicates that careers become sustainable when individuals have the opportunity to exercise their capabilities in order to live a dignified life. Nussbaum [16] defines ten central human capabilities, including the capabilities to live one's life, to have good health, bodily integrity, practical reason and relationships. Such capabilities would not have to be 'proactively' obtained or granted only when people show how 'flexible' they are, but these capabilities exist *because* of their integration into social justice concerns. This means, in line with Nussbaum [16], that capabilities function as fundamental entitlements, and that within society, people's capabilities should be respected and promoted in order to live a dignified life. Sustainable careers, therefore, should always be aimed at translating an entitlement to capability into practice and, as such, aim to promote social justice. In other words, careers must be designed to allow people to exercise their capabilities, or what they are effectively able to do and be [30]. This is important in the context of the current analysis, as a mere focus on individual proactivity and flexibility is exclusive for those people unable to do so. In contrast, by focusing on capabilities, the effects of dignity are twofold: on the one hand, the question becomes what and how people can contribute to organizational life and society such that they can exercise their capabilities, and express that which belongs to their personalities, skills, strengths, needs and so on. On the other hand, the question must be asked as to how people can exercise their capabilities such they are enabled to lead a dignified life, as careers should not be designed merely functionally to benefit organizations (or to ensure employees' 'happiness and health' in order to keep them productive), but also in line to enable them to exercise their capabilities in terms of their freedom in life. Dominant approaches in careers do not take inequalities, social injustice and the patriarchal, colonial and capitalist history and present into account. A combined dignity and capabilities approach puts this at the front of the organizing principle, and thereby raises the question:

*how can careers be designed in such a way that they become inclusively sustainable, such that people can sustainably exercise their capabilities?*

Such an approach prioritizes a focus on the more vulnerable people in and outside workplaces, with special emphasis on all those people across the world excluded from any chance to have a sustainable career. Hence, this explicitly normative conceptualization of sustainable careers aims for restorative social justice or the actually delivered justice for people across the world who have been marginalized in order to repair historical damage that has been enacted upon people across the world. Work, careers and capabilities are combined in a mutual understanding of their interactive nature: careers can only be sustained through meaningful work, as a sequence of jobs that have little meaningfulness attached to them are rarely considered as a 'career' [1]. In addition, to have a career of meaningful work experiences requires individuals to be able to exercise their capabilities in order to be a full human being. A dignity perspective dictates that such meaningful work sequences cannot be sustained through exploitation or the use or abuse of other people or the planet more widely. Put positively, dignity requires a collaborative approach to career building [13] and an emphasis on dignity protection and promotion of people and the planet. Notwithstanding such rather abstract notions of sustainable careers, the concern is how this can be translated into practice.

## 5. Practical Implications

A standard critique of dignity and capabilities theory is that it is either utopian or too abstract, and not grounded in reality. It is explicitly *normative* and thereby projects a norm upon society and workplaces about how social justice and dignity can be enhanced. When dignity and capabilities inform career frameworks and actual decision making, it is needed to postulate a way to translate these theoretical-abstract concepts into their materialization. One straightforward way to implement dignity into organizational life is through the 'dignity question' [13]; in every decision that needs to be made within organizational life, the question should be raised to what extent it violates, respects, protects or promotes the dignity of oneself, the team, others within and outside the organization, and the planet more widely. Usually, contemporary organizational life is dictated through paradoxes [31], or the choice between conflicting tensions that jointly constitute an impossible choice, such as the choice between profitability and ecological concern. A dignity paradigm provides a specific direction by postulating the dignity question: to what extent does a productivity focus violate or protect dignity? Analysis may elucidate that profit is generated at the *expense* of the planet (such as would be the case with fossil fuel companies), and therefore careers may become only *truly* sustainable if they take into account the broader notions of sustainability, dignity and capabilities.

Organizations are therefore advised to incorporate the dignity concern in all their practices, for instance through binding regulatory frameworks. Such regulation can also be enforced through governments, in the same vein as collective labor agreements apply to working conditions. This involves the embedding of organizational analysis on the externalities of its production processes, and the extent to which costs are externalized on people and the planet, while the primary benefits are maintained with the company (e.g., profits are generated and subsequently offshored to tax havens, or used to buy back company stocks). Careers alike must be designed with the dignity question in mind, and organizations and managers need to design career paths for employees that respect and protect the dignity of the employees, and others within and outside the organization.

This also necessitates the more fundamental rethinking of careers and positions within organizations: organizational hierarchies in which only a few of the 'best' employees can move upward and have better rewarded jobs are, by definition, exclusive. A better approach to such matters is to rethink hierarchy itself, and for instance to make positions of authority and responsibility by definition temporary, such that people are only in such positions for a defined period of time, after which others can take over their position (see also [13]. Organizational democracy is also a way through which democratic processes to

the development of careers can be integrated into organizational functioning [32]. However, democratizing organizations means to shift the power away from the top to be distributed across the organization, a process and transformation that will meet resistance from the top. Hence, governmental (top-down) and collective worker (bottom-up) pressure to democratize organizations may be of importance here, but only if democracy is combined with dignity, such that elections and democratic principles are aimed at the protection and promotion of dignity.

## 6. Conclusions

Sustainable careers have become hypernormalized ideology [13]. The concept operates as a fantasy that masks the more traumatic nature of the unsustainability of contemporary careers and working lives. Moreover, by projecting a perspective of the career as inherently able to become sustainable through individual action, lines of exclusion are ignored, and the fact that most people on this planet do not have access to a career at all can be bypassed [33]. This paper has analyzed why this is, and how a dignity and capabilities perspective may shed new light on how careers can be made more sustainable. There is a possibility for greater sustainability, but only when careers are linked to the dignity of other people and the planet more widely. Accordingly, a 'golden rule' can be formulated that careers can only be sustainable if they do not violate, and if they do respect and protect the dignity of others. Moreover, dignity can be achieved when people are able to exercise their capabilities to lead a dignified life. Hence for career counsellors and organizational decision makers, the message is straightforward: careers have to be designed to allow people to exercise their capabilities to lead their lives in (dignified) freedom and to provide them with the opportunity to be a full human being. This may provide the possibility for dignity at work, and more dignified careers. The social relevance of this paper includes the refocusing of a concept that has become overly used in neoliberal connotations towards an appreciation of the concept as potentially contributing to greater dignity of people and the planet. However, to do so, we need a much more critical engagement with the concepts we use in our society and rethink them radically such that we strive towards *real* sustainability, rather than using such terms merely to comply with dominant discourse in society, which ultimately fails to achieve real change and improvement of living conditions and our planet as well. In sum, this paper and essay is one of hope, as within the ruins of the contemporary broken society new perspectives can be formulated and found. When dignity and capabilities prevail, new avenues can be found for impactful research and practice of careers.

**Funding:** This research received no external funding.

**Institutional Review Board Statement:** Not applicable.

**Informed Consent Statement:** Not applicable.

**Data Availability Statement:** Data are contained within the article.

**Conflicts of Interest:** The authors declare no conflict of interest.

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
