# Peer review of "A Psychology of Sustainable Career Development: Hypernormalized Ideology or Inherently Sustainable?"

_sustainability, doi:10.3390/su16020578_

Round 1

Reviewer 1 Report

Comments and Suggestions for Authors

I truly recommend to clearly define the objective of the paper in the abstract. Also, include a section of social relevance about the review to clarify the justification about the publication. Some of the references in the text are not included in the list of references (Bal et al., 2023; Bal et al., 2021). I recommend to revise additional literature that could be included in the review as support of your conceptual proposal. 

Author Response

Reviewer 1

I truly recommend to clearly define the objective of the paper in the abstract. Also, include a section of social relevance about the review to clarify the justification about the publication. Some of the references in the text are not included in the list of references (Bal et al., 2023; Bal et al., 2021). I recommend to revise additional literature that could be included in the review as support of your conceptual proposal. 

Response:

Thanks! We have now added:

-the objective of the paper (see p2).

-Social relevance of the paper (p12).

-references to the right literatures. Some references were not updated, yet, hence the confusion. All is clear now.

-Added additional literature.

Thanks for your comments!

Reviewer 2 Report

Comments and Suggestions for Authors

Strengths: This article presents a very relevant and current topic with a relevant literature review on the topic.

Improvement proposals

Weaknesses: the summary must include (objectives, methodology and results). The main objective must be stated in the introduction, as well as the research questions. After the literature review, methodology, analysis of results and conclusions must be included.

Line 16 – “Anchored in an interpretation of sustainable careers as promoting dignity and capabilities of people, a new psychology of the sustainable career can be formulated. Authors must describe the methodology used in this study

Line 40 “The intellectual underdevelopment underlying such models requires a 40 one dimensional supportive empirical approach: cases and statistics are sought to support such models, whereby successful employees” The authors must concretely explain this statement.

Line 252 – „On the other hand, those who have experienced their ‘absurd moment’ (Bal et al., 2023) may see through the smokescreen of authoritative discourse“ Authors must explain what they understand by absurd moment.

Line 291 – „In contrast, this paper offers an understanding of sustainable careers as conceptualized through dignity and capabilities. The authors discuss how dignity and capabilities contribute to sustainable careers.

Linha 494 …”lines of exclusion are ignored, and the fact that most people on this planet do not have access to a career at all can be bypassed. Authors must substantiate this claim with data and concrete information.

The conclusions must include the value of this study for understanding the topic, limitations and future studies.

Author Response

Reviewer 2

Strengths: This article presents a very relevant and current topic with a relevant literature review on the topic.

Improvement proposals

Weaknesses: the summary must include (objectives, methodology and results). The main objective must be stated in the introduction, as well as the research questions. After the literature review, methodology, analysis of results and conclusions must be included.

Line 16 – “Anchored in an interpretation of sustainable careers as promoting dignity and capabilities of people, a new psychology of the sustainable career can be formulated. Authors must describe the methodology used in this study

Line 40 “The intellectual underdevelopment underlying such models requires a 40 one dimensional supportive empirical approach: cases and statistics are sought to support such models, whereby successful employees” The authors must concretely explain this statement.

Line 252 – „On the other hand, those who have experienced their ‘absurd moment’ (Bal et al., 2023) may see through the smokescreen of authoritative discourse“ Authors must explain what they understand by absurd moment.

Line 291 – „In contrast, this paper offers an understanding of sustainable careers as conceptualized through dignity and capabilities. The authors discuss how dignity and capabilities contribute to sustainable careers.

Linha 494 …”lines of exclusion are ignored, and the fact that most people on this planet do not have access to a career at all can be bypassed. Authors must substantiate this claim with data and concrete information.

The conclusions must include the value of this study for understanding the topic, limitations and future studies.

Response:

Thanks for your kind words! We have adapted the paper including the following changes:

-Included objectives for the paper in the introduction. As it is a conceptual paper, there are no methods, analyses and results, so these were not included.

-Line 16: we added the method (i.e., conceptual analysis) of this paper.

-Line 40: we have explained this statement.

-Line 252: thank you so much for this: we now explain what the absurd moment is, in line with the original conceptualization by Albert Camus.

-Line 291: thanks – in the section following this statement, we explain how dignity and capabilities can be theorized in relation to sustainable career development.

-line 494: we have now included appropriate referencing to evidence around this.

-As this concerns a conceptual review and essay, we chose not to adhere to the standard format for empirical paper, including a limitations and future research section. On the basis of the reading of the paper, we can imagine readers to make up their own impression of the work and the value of the paper.

Thank you for your suggestions!

Reviewer 3 Report

Comments and Suggestions for Authors

Good presentation of basic constructs of the review paper within the abstract, together with presentation of the main issues and with good, adequate choice of the key words. Generally, this is an interesting review article with the aim of better understanding of sustainable career development.

There is no typical section of literature review / there are a lot of examples and sources but it should be improve a little bit more / to put some articles connected with burnout syndrome and make some short review about connection between them: is there any dependency and if is how big and/or important is.

You could find a few suggestions for improving the paper:

Page 2, line 91.

It should not be referred just to Western Workers but also to workers of Western Balkan i.e. to those who are on the waiting list for EU – they are trying to reach EU system.

Same page, line 96.

Not only Western Universities. Why you putted just Western Universities?

To improve literature review you should check this paper

Su-Chiun Liang An-Tien Hsieh (2005) Individual's Perception of Career Development and Job Burnout Among Flight Attendants in Taiwan, The International Journal of Aviation Psychology, 15:2, 119-134, DOI: 10.1207/s15327108ijap1502_1

Generally very good paper and with some minimal changes could be publish.

Author Response

Reviewer 3:

Good presentation of basic constructs of the review paper within the abstract, together with presentation of the main issues and with good, adequate choice of the key words. Generally, this is an interesting review article with the aim of better understanding of sustainable career development.

There is no typical section of literature review / there are a lot of examples and sources but it should be improve a little bit more / to put some articles connected with burnout syndrome and make some short review about connection between them: is there any dependency and if is how big and/or important is.

You could find a few suggestions for improving the paper:

Page 2, line 91.

It should not be referred just to Western Workers but also to workers of Western Balkan i.e. to those who are on the waiting list for EU – they are trying to reach EU system.

Same page, line 96.

Not only Western Universities. Why you putted just Western Universities?

To improve literature review you should check this paper

Su-Chiun Liang & An-Tien Hsieh (2005) Individual's Perception of Career Development and Job Burnout Among Flight Attendants in Taiwan, The International Journal of Aviation Psychology, 15:2, 119-134, DOI: 10.1207/s15327108ijap1502_1

Generally very good paper and with some minimal changes could be publish.

Response:

Thank you! We have made the following changes to the paper.

-We have added links to burnout, which is also the countereffect of neoliberal career development systems which emphasize individual responsibility without proper organizational/governmental support, which relates to higher burnout.

-Line 91/96: thanks for these comments. We have updated these to reflect more accurately the situations of scholars researching sustainable careers. We spoke of ‘Western’ universities as most of the literature on sustainable careers is (still) conducted in the Western world, even though more research is now conducted elsewhere.

-Thanks for the reference.

Reviewer 4 Report

Comments and Suggestions for Authors

The article presents a significant issue, that of sustainable career, approaching it from a non-neoliberal point of view, closer to issues of social and environmental justice. I find this contribution and the carefully crafted critical analysis valuable.

Within the body of research on these issues, vocational guidance and career counseling more specifically, there are critical voices that emphasize some of the themes discussed and that highlight the need for critical consciousness and values centered on inclusion, social and the environment justice. The authors should take these into account for a more correct and comprehensive analysis

Author Response

Reviewer 4:

The article presents a significant issue, that of sustainable career, approaching it from a non-neoliberal point of view, closer to issues of social and environmental justice. I find this contribution and the carefully crafted critical analysis valuable.

Within the body of research on these issues, vocational guidance and career counseling more specifically, there are critical voices that emphasize some of the themes discussed and that highlight the need for critical consciousness and values centered on inclusion, social and the environment justice. The authors should take these into account for a more correct and comprehensive analysis

Response:

Thanks for your comments! Much appreciated. We now included these literatures as well in our revised paper.

Round 2

Reviewer 2 Report

Comments and Suggestions for Authors

The authors added the requested observations and suggestions.